# Association between long-term use of calcium channel blockers (CCB) and the risk of breast cancer: a retrospective longitudinal observational study protocol

Chau Ho ,[1] Ninh Thi Ha,[1] David Youens ,[1,2] Walter P Abhayaratna,[3,4] Max K Bulsara ,[5] Jeffery David Hughes ,[6,7] Gita Mishra,[8] Sallie-Anne Pearson,[9,10] David B Preen,[10,11] Christopher M Reid,[1] Rikje Ruiter,[12,13] Christobel M Saunders,[14,15,16] Bruno H Stricker ,[12] Frank J A van Rooij,[12] Cameron Wright,[1,14,17,18] Rachael Moorin [1,11]

For numbered affiliations see end of article.

**Correspondence to**
David Youens;
david.youens@curtin.edu.au

## ABSTRACT

**Introduction** Calcium channel blockers (CCB), a commonly prescribed antihypertensive (AHT) medicine, may be associated with increased risk of breast cancer. The proposed study aims to examine whether long-term CCB use is associated with the development of breast cancer and to characterise the dose–response nature of any identified association, to inform future hypertension management.

**Methods and analysis** The study will use data from 2 of Australia's largest cohort studies; the Australian Longitudinal Study on Women's Health, and the 45 and Up Study, combined with the Rotterdam Study. Eligible women will be those with diagnosed hypertension, no history of breast cancer and no prior CCB use at start of follow-up (2004–2009). Cumulative dose-duration exposure to CCB and other AHT medicines will be captured at the earliest date of: the outcome (a diagnosis of invasive breast cancer); a competing risk event (eg, bilateral mastectomy without a diagnosis of breast cancer, death prior to any diagnosis of breast cancer) or end of follow-up (censoring event). Fine and Gray competing risks regression will be used to assess the association between CCB use and development of breast cancer using a generalised propensity score to adjust for baseline covariates. Time-varying covariates related to interaction with health services will also be included in the model. Data will be harmonised across cohorts to achieve identical protocols and a two-step random effects individual patient-level meta-analysis will be used.

**Ethics and dissemination** Ethical approval was obtained from the following Human research Ethics Committees: Curtin University (ref No. HRE2022-0335), NSW Population and Health Services Research Ethics Committee (2022/ETH01392/2022.31), ACT Research Ethics and Governance Office approval under National Mutual Acceptance for multijurisdictional data linkage research (2022.STE.00208). Results of the proposed study will be published in high-impact journals and presented at key scientific meetings.

## STRENGTHS AND LIMITATIONS OF THIS STUDY

⇒ This study will characterise any dose–response nature of the association between calcium channel blockers (CCB) use and breast cancer.
⇒ To minimise the impact of confounding by indication, the study will be limited to participants with diagnosed/self-reported hypertension and comparisons will include participants exposed to other prescribed (non-CCB) antihypertensive medicines in addition to those who have no pharmacotherapy for hypertension.
⇒ Complete ascertainment of prior exposure will not be possible since we will not have lifetime capture of prescribing data; therefore, CCB exposure may be under-reported.
⇒ The diagnostic interval of breast cancer may vary between Australian and Dutch women, particularly since case ascertainment through cancer registries can take months from the initial diagnosis to coding notifications.

**Trial registration number** NCT05972785.

## INTRODUCTION

Breast cancer has overtaken lung cancer as the most common cancer and the leading cause of death in women globally, accounting for 2 million new cases and 685 000 deaths worldwide in 2020.[1] By 2040, breast cancer is expected to increase by 40% to more than 3 million new cases and 1 million deaths annually.[2] Breast cancer places a significant burden of disease, excess mortality and reduced quality of life on women and the broader health system; identifying and mitigating

BMJ

potentially modifiable risk factors are a critical strategy in reducing this burden.

Hypertension is a well-established, modifiable risk factor for stroke and other cardiovascular events and a major contributor to chronic diseases such as heart failure, chronic kidney disease and retinopathy.[3] Worldwide, hypertension causes more deaths and morbidity than any other biomedical risk factor.[4] Cardiovascular disease is responsible for approximately 17 million deaths a year, nearly one-third of total mortality globally.[5] Together with three other AHT medicines (ie, ACE inhibitors (ACEI), angiotensin receptor blockers (ARB) and thiazide or thiazide-like diuretics), dihydropyridine calcium channel blockers (CCBs) have been recommended for initiation as first-line therapy in many countries.[6 7]

It has been hypothesised that CCBs may increase the risk of cancer by changing intracellular calcium levels, thereby affecting the process of apoptosis (programmed cell death), disabling the destruction of mutated cells and leading to development of diseases such as cancer in which there is indiscriminate replication of a mutated cell.[8] Calcium is known to play a regulatory role in apoptosis via multiple biochemical/pharmacological pathways.[9–11] It has been postulated that, due to its secretory nature, breast tissue may be more vulnerable to alterations in apoptotic activity than other tissue, leading to complex hormone-related relationships between apoptosis and breast carcinoma.[12] In addition, one particular CCB, nifedipine, has been found to increase proliferation and migration of breast cancer cells in vitro[13] and to reduce resting calcium concentration and apoptotic gene expression in mice.[14]

In 2017, we published a systematic review and meta-analysis[15] highlighting the uncertainty regarding the association between long-term use of CCBs and breast cancer, primarily due to the limited availability of long-term follow-up data. This review and meta-analysis found that the literature is equivocal on whether CCBs are a risk factor for breast cancer. Li et al[16] reported that long-term (>10 years) use of CCBs was associated with a 2.4-fold (95% CI 1.2 to 4.9) increased risk of invasive ductal breast cancer and 2.6-fold (95% CI 1.3 to 5.3) increased risk of invasive lobular breast cancer. In contrast, Grimaldi-Bensouda et al,[17] using data from the UK, and Wilson et al,[18] using data from the USA, from sisters with and without breast cancer failed to find an association between CCB use and breast cancer (OR 0.95 (95% CI 0.87 to 1.04) and HR 0.88 (95% CI 0.58 to 1.33), respectively). A similar pattern was observed in a recent study conducted by Rotschild et al (2022) in Israel, revealing an OR 0.997 (95% CI 0.962 to 1.034) for a long-term exposure to CCBs (above 8 years) and breast cancer risk.[19] A non-significant association was found in US women aged 55+ years for CCB use but a protective effect for ACEI use.[20] However, a study by Gómez-Acebo et al[21] using Spanish data found CCB use for 5 or more years was associated with a 1.77 times increased risk (95% CI 0.99 to 3.17) of breast cancer. A 2021 systematic review of clinical trials showed no strong effect of any type of AHT medicine on breast cancer with a median follow-up of 4.2 years. However, a systematic review of cohort studies with longer follow-up (≥10 years) published in the same year found that the use of beta blockers, CCBs and diuretics was associated with increased breast cancer risk.[22] The authors noted the heterogeneity of study design, study duration and capture of drug exposure status and also that most studies were from Europe or North America, limiting external validity.

Recommendations for CCB use are different across countries. CCBs are currently the second most commonly prescribed AHT medicine and are considered first-line treatment for hypertension in Australia.[23] In contrast, CCBs are not recommended for first-line therapy in the Netherlands, where thiazide diuretics are more commonly initially prescribed for blood pressure control[24] and this represents an opportunity to investigate the role of prescribing practices in cancer risk. Variations in prescribing patterns may drive the differences in the associations observed in studies of CCB use and risk of breast cancer across populations in addition to differences in underlying cancer risk due to varying environmental or biological/genetic factors (eg, body mass index (BMI), physical activity, smoking and alcohol history, hormonal medicine use or mammographic density). We expect this to be true for CCB prescribing in Australia compared with The Netherlands. The similarities and differences between the Dutch and Australian cohorts have been leveraged previously in studies of medicine safety.[25] We therefore propose to use data from these two countries to (1) explore confounding and (2) effect modification by these specific factors in the relationship between CCB use and breast cancer risk.

Long-term evaluation of medicine safety is best monitored via longitudinal pharmacovigilance and observational studies, as they have both linked claims data and good confounder adjustment obtained through surveys of participants whereas randomised clinical trials with sufficient follow-up are largely not feasible due to time and cost factors. Thus, we aim to examine whether long-term CCB use is associated with the development of breast cancer among women enrolled in three longitudinal cohort studies in Australia and The Netherlands and to characterise the dose–response nature of any such association. A secondary aim is to assess whether any differences in the association between CCB use and the development of breast cancer exist between Australian and Dutch women and, if so, to explore confounding and effect modification for a range of socioeconomic and clinical factors.

## METHODS AND ANALYSIS
### Study design
This will be a retrospective, observational study conducted across three internationally renowned longitudinal cohorts incorporating both self-report and administrative data linked at the person level. For the primary aim,

the study will use a distributed methodological approach, which has been shown to assure methodological quality and transparency and increase the likelihood that differences between results are due to true variation in effect size, rather than design or process-related problems.[26] Our analysis will (1) first use identical protocols across cohorts (harmonised analyses) and then (2) use variable/cohort-specific protocols to allow the influence of cohort-specific variables to be investigated (non-harmonised analyses).

## Data sources

We will use data from the Australian Longitudinal Study on Women's Health (ALSWH),[27] the New South Wales (NSW) Sax Institute's 45 and Up Study (45 and Up Study)[28] and the Rotterdam Study.[29]

► The ALSWH[27] is a national study that began in 1996 with a random sample of more than 40 000 women in 3 birth cohorts. The surveys were conducted every 3 years since 1996; however, the 1921–1926 cohort was surveyed 6 monthly from November 2011 for health and lifestyle changes. In our study, we will use two longitudinal cohorts, the 1946–1951 cohort (aged 45–50 years at recruitment) and the 1921–1926 cohort (aged 70–75 years at recruitment).

► Administrative data for the ALSWH are included for all Australian states and territories, linked using probabilistic linkage with clerical review by the data linkage units in each state. Linked data used for this study include hospital admissions, emergency department presentations, cancer registry and the national death index data. Hospitalisation and cancer registry data are recorded separately in each Australian state; hence, the start and end of these datasets vary according to the availability of the data in each state. Medicare Benefits Schedule (MBS) claims and Pharmaceutical Benefits Scheme (PBS) data were provided by Services Australia.

► The 45 and Up Study[28] is a longitudinal cohort of 267 357 participants aged ≥45 years residing in NSW, Australia at recruitment randomly sampled from the Services Australia Medicare enrolment database. Participants from rural and remote areas and those 80+ years of age were oversampled. Participants were recruited from 2005 to 2009 (2005 being a pilot study) and followed up every 5 years for changes in health and lifestyle. The response fraction was approximately 19%. Only female participants (53.6% of the cohort) recruited in the main study between 2006 and 2009 will be included in the proposed study.

► Survey data for the 45 and Up Study were linked to the NSW Admitted Patient Data Collection and the Australian Capital Territory (ACT) Admitted Patient Collection, the NSW and ACT Emergency Department Data Collection, the NSW and ACT Cancer Registry, the NSW and ACT Registry of Births Deaths & Marriages death registrations, the Australian Coordinating Registry Cause of Death Unit Record File, NSW and ACT BreastScreen and the NSW Pap Test Registry by the Centre for Health Record Linkage using a probabilistic linkage procedure with clerical review. The current estimated rate of false-positive linkages is 0.5% (www.cherel.org.au). Linkage of 45 and Up Study cohort data to Medicare claims and PBS data provided by Services Australia was facilitated by the Sax Institute using a unique identifier and deterministic matching.

► The Rotterdam Study[29] is a prospective cohort study of persons aged ≥45 years living in the city of Rotterdam, The Netherlands. The study started with a pilot phase in the second half of 1989, called 'RS-I'. A second cohort was recruited in 2000 (RS-II) and a third cohort in 2006 (RS-III). By the end of 2008, the Rotterdam Study comprised 14 926 participants aged 45 years or over. At baseline, participants were interviewed and had a clinical examination that was repeated every 3–4 years. Follow-up of participants between clinical examinations is undertaken via automated coupling of the study database with medical records from general practitioners (GPs), who receive all relevant medical information from all caregivers of their patients. The data include hospital and specialist discharge summaries providing dates of diagnoses and major surgical interventions. Only female participants of the three longitudinal cohorts will be included in our study.

► The three Rotterdam longitudinal cohorts are linked at the individual level to records for dispensed medicines from all pharmacies in the study area (1991–2018), the Netherlands Comprehensive Cancer Organisation (1989–2016) and municipality (death) records (1989–2023).

A summary of the source of the major categories of data available for the study is provided in table 1.

## Study population

The same study population will be used for the primary and secondary aims. Entry to our study will be on survey completion date for women who participated in the ALSWH survey waves between 2004 and 2008 or the 45 and Up Study baseline survey between 2006 and 2009 or the date of the Rotterdam Study interview and clinical examinations for women who completed these between 2004 and 2008. Inclusion criteria will be women who at entry to our study (1) were alive and consented to linkage of administrative data, (2) have evidence of hypertension from either self-reported hypertension (Australian longitudinal cohorts) or have a recorded diagnosis of hypertension in clinical examination data (Rotterdam cohorts).

Women will be excluded at entry if they had (1) history of primary invasive/in situ breast cancer or cancer that is metastatic to the breast; or (2) history of bilateral mastectomy; or (3) evidence of CCB use prior to entry. Further exclusion criteria will be applied in Australian longitudinal cohorts only as per standard practice when using PBS data: (1) women with any evidence of being

**Table 1** Sources of data for the study

| Data source | Sociodemographic characteristics | Lifestyle characteristics | Clinical characteristics | Reproductive health | Family history of cancer and genetic predisposition | Healthcare utilisation | Diagnosis of breast cancer | Medication use | Death |
|---|---|---|---|---|---|---|---|---|---|
| Self-report survey/clinical examinations | x | x | x | x | x | | x | x | |
| PBS/pharmacy data | | | x | | | | | x | |
| Cancer Registry data | | | x | | | | x | | |
| MBS data | | | x | | | x | | | |
| GP records | | | x | | | x | x | | |
| Hospital admission data | | | x | | | x | x | | |
| Mortality register data | | | | | | | | | x |

PBS, Pharmaceutical Benefits Scheme; MBS, Medicare Benefits Schedule; GP, general practitioner.

Department of Veterans' Affairs (DVA) concession card holders, because DVA-subsidised prescriptions are funded separately and not captured in the PBS data held; or (2) women who did not have evidence of concessional healthcare status for more than 1 year during period from 2003 to 2012. Those with concessional status, for example, those receiving the aged care pension or certain other government benefits, have lower copayments for PBS-listed medicines in comparison to 'general beneficiaries'. This exclusion is necessary because some AHT medicines fall below the PBS copayment threshold for general beneficiaries, hence attracted no government reimbursement and were not captured in PBS data prior to an administrative change in 2012.[30] In contrast, the lower copayment threshold for concessional patients means that AHT medicines can be captured throughout the study period.

Participants will be followed until the earliest date of one of the following: the outcome (a diagnosis of invasive breast cancer); a competing risk event (evidence of bilateral mastectomy without a diagnosis of breast cancer, death prior to any diagnosis of breast cancer) or end of the study follow-up (censoring event), whichever came first. Note that since the duration of the available data differs across the longitudinal cohorts, end of study follow-up will be established for each person based on the data available. Figure 1 presents a schematic diagram of timeline for entry, exposure and study exit (ie, recording of the outcome, censoring, competing event or end of follow-up) of our study population.

As administrative data are available prior to entry, this allows for a 'lookback' period to ascertain some aspects of health service use prior to study entry, for comorbidity ascertainment and to remove those with a history of CCB use or breast cancer diagnosis prior to time-zero for follow-up.

### Harmonisation of data

Harmonisation of data will be undertaken using the six-step process described by Rolland *et al*[31] and Fortier *et al*.[32] After obtaining approvals from all data access and ethics committees, the Australian data will be housed in cohort-specific folders in the Sax Institute's Secure Unified Research Environment (https://www.saxinstitute.org.au/solutions/sure/), whereas the Rotterdam data will be uploaded and analysed in the Accessible Network Digital Research Environment Alliance (www.andrea-cloud.eu). Related documents such as data processing scripts and comments on specific decisions taken throughout the process will be recorded. An associated data dictionary will also be developed. To ensure content equivalence, each variable will be checked on (1) the definition used in the questionnaire, format, categories, unit and time frame, (2) measurement method (eg, self-reported, clinical examination, etc) and (3) harmonisation rules. Variables relating to the main exposure, outcomes/competing risk, sociodemographic, lifestyle, clinical, reproductive characteristics and healthcare utilisation that are harmonisable, together with those available for the cohort-specific

**Figure 1** Schematic diagram showing entry, exposure and exit of the study populations across longitudinal cohorts. CCB, calcium channel blockers.

analyses, are presented in table 2. Where a variable has different categories across cohorts, we will collapse categories to generate the closest possible match.

### Exposure ascertainment

The exposure in this study, for both the primary and secondary aims, will be CCB dispensing including both combination and monotherapy products. Exploring the impact of confounding by indication for non-CCB AHT medicines is important because a modest increased risk of cancer in individuals with hypertension has been documented.[33] To avoid this issue, previous studies have used a comparator group of patients exposed to other AHTs and/or with a hypertension diagnosis.[34] Our study population is defined by the presence of hypertension since comparing only to a reference AHT medicine group risks biasing towards the null, if the comparator drug is associated with the outcome. Thus, to reduce bias to the null, women who are both AHT and CCB naïve will be included in the study as long as they meet the entry criteria of hypertension at baseline as described previously. To ensure that exposure to other AHT medicines is accounted for, the cumulative dose-duration exposure of other prescribed AHT medicines (combined) will be captured in the same way as that described for CCB exposure. Thus, in our study, the following four exposure states will be possible during follow-up: (1) women with no exposure to either CCBs or other AHTs; (2) women exposed to other AHTs but not exposed to CCBs; (3) women exposed to CCBs but not exposed to other AHTs and (4) women exposed (at any time) to both CCBs and AHTs. To compensate for immortal-time bias, and to be able to capture the dose–response nature of the association of CCB exposure with risk of breast cancer, a time-varying continuous ascertainment of cumulative exposure (dose duration) to CCB (and separately other AHTs) will be captured as follows.[35]

We will use the Anatomical Therapeutic Chemical (ATC) codes[36] and information on dispensed medicines in the PBS data for Australian longitudinal cohorts and medicine record data for the Rotterdam longitudinal cohorts to identify AHT drugs dispensed for the study population from study entry to exit. We will categorise AHT drugs dispensed into five main AHT drug classes: (1) beta-blocking agents (ATC code C07), (2) CCB (C08); (3) diuretics (C03), (4) agents acting on the renin–angiotensin system (ACEIs and ARBs) (C09), or (5) other AHTs not included in these classes regardless of treatment therapies (ie, monotherapy or combination therapy). Details of level 5 ATC codes of the dispensing medicines are presented in online supplemental appendix table 1.

For all longitudinal cohorts, the total cumulative duration exposed and the total cumulative dose of each AHT drug class will be calculated from the medicine records available from entry to exit of the study. For all cohorts, the data include information on the date of dispensing, quantity dispensed and the strength of the medicine (derived from the PBS item code for Australian data). In the Rotterdam data, the daily prescribed dose and prescribed duration of use are also available.[29] This information is not present in the Australian data necessitating estimation of the duration of the medicine. For the Australian data, the reverse waiting time distribution method with multiple random index dates will be used to determine the following: (1) the median duration of medicine possession, defined as the number of days that 50% of prevalent users of the same medicine refill their scripts and (2) the upper duration limit of medicine possession, defined as the number of days that 80% of prevalent users of the same medicine refill their scripts.[37 38] The median duration will be used as the usual possession duration of the medicine in calculating the total duration (days) of medicine possession in contiguous periods of exposure. The contiguous period will be defined as a series of dispensings of the same AHT class with no break between dispensing that is greater than the upper duration limit of medicine possession of the last medicine in the series. The total cumulative duration (days) of exposure to each

**Table 2** Variables for harmonised and cohort-specific analyses

| | ALSWH | | 45 and Up Study | Rotterdam study | | | Harmonisable status |
|---|---|---|---|---|---|---|---|
| | 1946–1951 Cohort | 1921–1926 Cohort | | RS-I-4 | RS-II-2 | RS-III-1 | |
| **Sociodemographic characteristics** | | | | | | | |
| Age | Y | Y | Y | Y | Y | Y | Y* |
| Marital status | Y | Y | Y | Y | Y | Y | Y* |
| Country of birth | Y | Y | Y | N | N | N | N |
| Socioeconomic disadvantage | Y | Y | Y | Y | Y | Y | Y* |
| Remoteness | Y | Y | Y | NA | NA | NA | Y* |
| Work status | N | N | Y | Y | Y | Y | N |
| Education status | Y | Y | Y | Y | Y | Y | Y* |
| **Lifestyle characteristics** | | | | | | | |
| Body mass index (height/weight) | Y | Y | Y | Y | Y | Y | Y* |
| Physical activity | Y | Y | Y | N | N | Y | N |
| Smoking status | Y | N | Y | Y | Y | Y | N |
| Alcohol drinks per week | Y | N | Y | Y | Y | Y | N |
| Shift work at night | Y | N | N | N | N | N | N |
| **Clinical characteristics** | | | | | | | |
| High BP | Y | Y | Y | Y | Y | Y | Y* |
| Systolic BP/diastolic BP | N | N | N | Y | Y | Y | N |
| Blood cholesterol level | N | N | N | Y | Y | Y | N |
| Diabetes | Y | Y | Y | Y | Y | Y | Y* |
| Heart disease | Y | Y | Y | Y | Y | Y | Y* |
| Stroke | Y | Y | Y | Y | Y | Y | Y* |
| Family history of heart disease | N | N | Y | Y | Y | Y | N |
| Family history of stroke | N | N | Y | Y | Y | Y | N |
| Mastectomy | Y | Y | Y | Y | Y | Y | Y* |
| **Reproductive health** | | | | | | | |
| Age at menarche | Y | N | N | Y | Y | Y | N |
| Age at menopause | Y | N | Y | Y | Y | Y | N |
| Age when had first child | Y | N | Y | Y | Y | Y | N |
| Number of children | Y | Y | Y | Y | Y | Y | Y* |
| History of hysterectomy/both ovaries removed | Y | Y | Y | Y | Y | Y | Y* |
| Breast feeding | N | N | Y | Y | Y | N | N |
| **Healthcare utilisation** | | | | | | | |
| Concession status | Y | Y | Y | NA | NA | NA | Y* |
| Mammogram | Y | N | Y | N | N | N | N |

Continued

**Table 2** Continued

| | ALSWH | | 45 and Up Study | Rotterdam study | | | Harmonisable status |
|---|---|---|---|---|---|---|---|
| | 1946–1951 Cohort | 1921–1926 Cohort | | RS-I-4 | RS-II-2 | RS-III-1 | |
| Hospital admissions | Y | Y | Y | N | N | N | N |
| Specialist/consulting physician attendances | Y | Y | Y | N | N | N | N |
| GPs attendances | Y | Y | Y | N | N | N | N |
| Diagnosis of cancer | | | | | | | |
| A diagnosis/history of invasive breast cancer | Y | Y | Y | Y | Y | Y | Y* |
| A diagnosis/history of non-invasive breast cancer (LCIS, DCIS) | Y | Y | Y | Y | Y | Y | Y* |
| Other cancer | Y | Y | Y | Y | Y | Y | Y* |
| Medicine use | | | | | | | |
| CCB | Y | Y | Y | Y | Y | Y | Y* |
| Non-CCB (ACEI, ARB, BB, diuretics, others) | Y | Y | Y | Y | Y | Y | Y* |
| Statins | Y | Y | Y | Y | Y | Y | Y* |
| Current use of combined hormonal contraception | Y | Y | Y | Y | Y | Y | Y* |
| Current use of combined menopausal hormone therapy | Y | Y | Y | Y | Y | Y | Y* |
| Comorbidity (Rx-Risk) score | Y | Y | Y | Y | Y | Y | Y* |
| Mortality | | | | | | | |
| Death | Y | Y | Y | Y | Y | Y | Y* |

This table includes all potential confounders available from ALSWH (https://alswh.org.au/for-data-users/data-documentation/data-dictionary/), 45 and Up Study (https://www.saxinstitute.org.au/solutions/45-and-up-study/use-the-45-and-up-study/data-and-technical-information/), and Rotterdam Study (https://www.erasmusmc.nl/en/research/core-facilities/ergo-the-rotterdam-study).
*These variables will be used in the harmonised analysis.
†Comorbidity (Rx-Risk) score includes 46 clinical conditions based on prescribed medicine claims data.
ACEI, ACE inhibitors; ALSWH, Australian Longitudinal Study on Women's Health; ARB, angiotensin receptor blockers; BB, beta blockers; BP, blood pressure; CCB, calcium channel blockers; CVD, cardiovascular disease; DCIS, ductal carcinoma in situ; FNA, fine-needle aspiration; LCIS, lobular carcinoma in situ; N, no data available; NA, not applicable; Y, Yes, data available.

AHT class under investigation for each individual over the study period will be the sum of the number of days accrued over all contiguous periods of exposure. The total cumulative dose will be similarly captured across all contiguous periods of exposure by summing the total milligrams (mg) of medicine dispensed of all dispensations (strength/unit multiplied by total units dispensed).

Contiguous periods of exposure will account for any carryover of oversupply. The carryover will only be applied if the next dispensing is the same medicine (same ATC code at the fifth level) as suggested previously.[39] The start of contiguous periods will be deemed the date of first-time dispensing observed within a series. End of a contiguous period will be the last dispensing date in the series plus the usual duration of medicine possession and any eligible carryover or the date of exit of the study where the duration of medicine possession continues past this date. Details of cumulative dose and duration are presented in figure 2.

In addition, time since last CCB exposure will be calculated in months from exit of the cohort back to the date of the last CCB exposure to differentiate between recent and previous exposure. This value will range from 0 for those with current exposure at the time of exit to the total of the follow-up time for those who are never exposed.

In the statistical model, dose duration will be captured by three elements, total cumulative duration, total

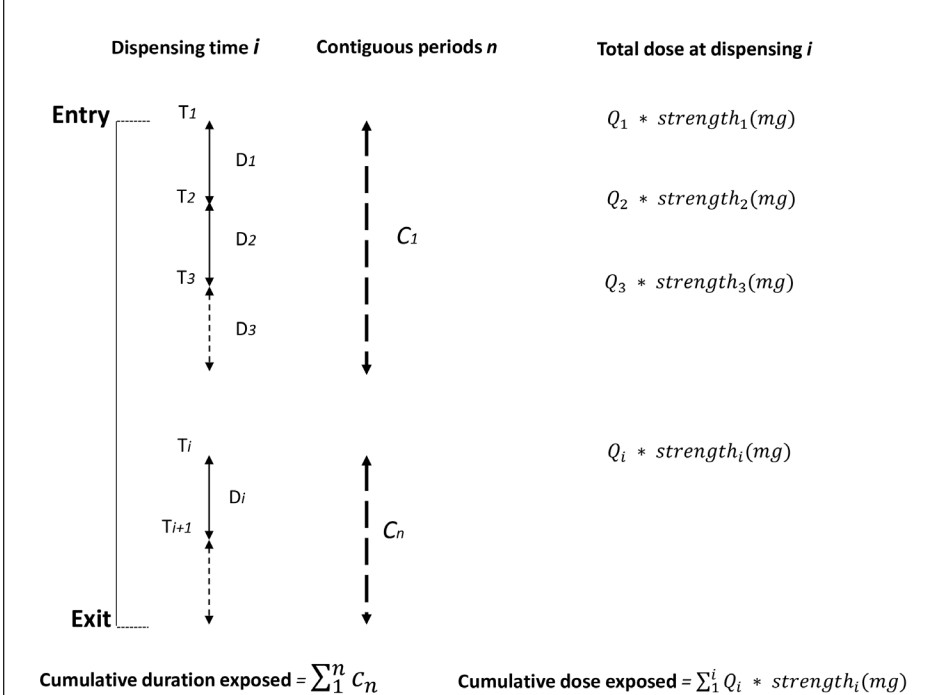

**Figure 2** Capture cumulative dose and duration exposed for each antihypertensive medicine.

cumulative dose and an interaction term to evaluate effect modification between duration and dose.

## Outcome ascertainment

During follow-up, incident diagnoses of invasive breast cancer will be identified using the International Classification of Diseases (ICD) version 10 Australian Modification[40] for the Australian cohorts and the ICD-10[41] for the Rotterdam Study. ICD-10 code C50.x (malignant neoplasm of breast) will be used to identify invasive breast cancer diagnosis across cancer registries and hospital admission data. In situ breast cancer (eg, D05.x) will not be included in the outcome because hyperplasia and atypical lesions are not precancerous (although some are risk factors), and only some ductal carcinoma in situ are obligate precursors.[42 43]

Cancer registries for all cohorts contain, by statutory requirement, records of all cancers diagnosed or treated and are therefore considered internationally the 'gold standard' for cancer identification for research purposes. Since cancer diagnoses for Dutch women and cancer registry data for the Australian women are only currently available until 2016 and 2018 respectively, breast cancers diagnosed later than these dates will be identified from GP data (Dutch women) and hospital data (Australian women). Hospital-derived diagnoses for breast cancer have a positive predictive value and sensitivity of 86% and specificity of 99.9% compared with the cancer registry in the 45 and Up Study population.[44] Sensitivity analyses showed that 53% of women with a cancer registry diagnosis had an inpatient diagnosis within the same calendar month and 93% within one calendar month; thus,

estimations of date of diagnosis align well with Cancer Registry dates.

The date of breast cancer diagnosis used in the study will be the date of diagnosis recorded in either the Australian Cancer Registry or the Rotterdam Study, or the date of the first hospital admission with a principal diagnosis of breast cancer/date recorded in the GP data where registry/study data are unavailable.

## Covariates

A wide range of sociodemographic, lifestyle and clinical potential confounders that may impact prescribing decisions for AHT medicines, a participant's decision to take prescribed AHT medicines or be associated with an increased risk of breast cancer will be included in the analysis.[45] Variables will include the following captured at baseline: age, marital status, highest qualification, socioeconomic disadvantage, remoteness, BMI, diabetes, heart disease, stroke, diagnosis of other cancer, comorbidity risk score, number of children and history of hysterectomy/both ovaries removed. In addition, the following time-varying factors (updated annually throughout follow-up and at exit) will be included: statin use, use of combined hormonal contraception, use of combined hormonal replacement therapy. Confounding by comorbidity will be assessed at baseline and also as a time-varying covariate as per the Rx-Risk score (with 5 years look back) that includes 46 clinical conditions based on ATC codes from prescribed medicine claims data, excluding AHT drugs as these medicines are captured in the exposure.[46] For the secondary aim (cohort-specific analyses), additional covariates will be available depending on the cohort; for

example, for the 45 and Up Study cohort, mammographic screening information will be used to differentiate screen-detected cancers; in the ALSWH and Rotterdam cohorts, multiple waves of self-reported/clinical examination data will be used to update lifestyle information during follow-up. In addition, where possible, we will explore the use of earlier study entry to provide longer follow-up. For both Australian cohorts hospital admissions, all specialist services, and GPs visits will be used as time-varying covariates to account for changing propensity to interact with health services. Differences between the cohort-specific analysis and the harmonised analysis undertaken for each cohort will be evaluated to aid with determination of the influence of cohort-specific variables on the association.

Details of the relevant codes (MBS item numbers (Australian women), ATC codes and ICD codes) used to identify participants' medical conditions, prescribed medicines and health service use from MBS, PBS/pharmacy data, hospital admission, GP data, cancer registries and mortality data collections are provided in online supplemental appendix table 2. A description of data items, questions and response categories from the baseline surveys of the ALSWH and the 45 and Up Study is presented in Online supplemental appendix table 3.

## Statistical analysis

*Primary aim:* for the primary aim, analysis of the association between CCB use and the development of breast cancer will be undertaken using Fine and Gray competing risks regression with incident breast cancer as the principal event, and death (from causes other than breast cancer) or bilateral mastectomy as the competing risk.[47] In this setting, the appropriate estimate of the probability of breast cancer is described by the cumulative incidence function rather than an HR. The models will account for confounding factors at entry using propensity scores to balance these factors across exposure status captured over the follow-up period. In addition, we will use standard covariate adjustment for time-varying covariates and any covariates that cannot be balanced using the propensity score. Any significant shifts in practice will be identified via the literature (and our clinical collaborators) and included as categorical variables in the models where appropriate.

Since the CCB exposure variable will be continuous rather than binary a generalisation of the binary propensity score, the generalised propensity score (GPS) developed by Hirano and Imbens will be used.[48] Since CCB exposure will be sporadic, we will also evaluate the dose-duration response effect by further extending the multivariable models to incorporate a time-dependent dose–response component.[49] Specifically, the dose-duration response function will be estimated after adjusting for covariate imbalance by using the GPS methods applied in previous publications.[49–52] The non-linear predictive relationships between the dose duration of CCB exposure and breast cancer will be modelled using non-linear threshold models developed by Gannon *et al*.[53] Threshold models allow for the differential effect of the dose duration of CCB exposure on the incidence of breast cancer with respect to an individual's duration and cumulative exposure to CCBs. The model will allow the coefficient ($\beta$) to vary according to the days/months/years per unit of the cumulative dose duration while assuming that other components in the model are constant across all individuals to identify the thresholds for subpopulations.[54] The estimation of the threshold models involves searching all possible values of the dose-duration function with maximum likelihood estimators computed for every tried value. For practical purposes, the number of subpopulations will be limited to 3 or less.[54] The threshold model that minimises the Bayesian information criterion used for selection of the non-linear models will be the final model. The counterfactual, individuals with no CCB exposure, will also be analysed as a baseline to show the impact of no versus any treatment using standard binary propensity score matching of those exposed versus non-exposed.[51]

*Secondary aim:* for the secondary aim, we will explore relationships between dose duration of CCB use and incident breast cancer and factors that confound and modify these relationships. Using multilevel modelling techniques and time-varying covariates, we will quantify the contributions of person-related, temporal-related and geospatial level-related factors to variation in the dose-duration effect. This will allow the development of both intercountry-specific and intracountry-specific profiles of women who are at higher and lower risk of CCB-attributable breast cancer.

Analyses will be adjusted for multiple comparisons where appropriate. Where the sample size is insufficient to confirm the nature of the association, the results will be regarded as exploratory. Analyses will use Stata SE V.15.[55] The analysis is scheduled to commence in February 2024 and conclude in July 2025.

## Subgroup analysis

In addition to our analysis of overall CCB exposure, the analysis for each aim will be conducted separately according to CCB class (ie, dihydropyridines and non-dihydropyridines) and duration of action (ie, short or long action) where sufficient statistical power exists. Moreover, we will conduct a subgroup analysis to assess the influence of a personal history of any cancer at cohort entry on the risk of breast cancer and CCB exposure.

## Sensitivity analyses

We will conduct several sensitivity analyses. Since hyperplasia and atypical lesions are not precancerous, we will test the effect of adding ductal carcinoma in situ as a competing risk in sensitivity analyses rather than as an exclusion criterion. We will explore the impact of imposing various time lags, based on clinical consultation, between the start of follow-up on the outcome to account for latency and plausible timing between exposure and outcome development. The potential impact of reduced exposure resulting from imperfect adherence

and prescribing at defined daily dose levels (a statistical measure of drug consumption, defined by the WHO) will be investigated in the dose-duration models and through sensitivity analyses.[36] We will expand the study population to women with prior CCB use at entry (captured as a starting dose-duration value using available look-back data) to examine the impact of CCB use (overall) on the outcome. Finally, because concessional status for subsidised medicines in Australia can change with age or employment status, we will explore varying the definition of continuous PBS concessional status in Australian women to determine the impact this has on the association of CCB exposure with the outcome. The main analysis for the study will be evidence of 100% concessional status between 2003 and 2012 with a single year of no evidence allowed (ie, at least one dispensation per year, and patient recorded as being on concessional status on each dispensation, with 1 year allowed to vary). This will be varied allowing 2 years of no evidence (the more relaxed scenario) and requiring 100% concessional status across all years (as the most stringent requirement). Sensitivity analyses will be applied to the primary aim in the first instance, and those factors found to be influential will be additionally tested in the secondary aim. Due to the potential duplication of participants who were part of the ALSWH and also enrolled in the 45 and Up Study residing in NSW, we will systematically exclude ALSWH participants residint in NSW in a sensitivity analysis. This exclusion is essential for evaluating the potential impact of duplication on the study results.

### Sample size calculation
Breast cancer is a relatively common event for the studies under investigation; thus, our sample size will be based on the women exposed to CCBs (as indicated by the numbers on the right-hand side of figure 3). Using a conservative ratio of unexposed:exposed of 5:1, we have 95% confidence with 80% power to detect a minimum effect size of ±0.2 for breast cancer associated with CCB use in all datasets given the number of exposed individuals.

### Individual patient data meta-analysis
For the primary aim, a two-step (since the data cannot be brought together) random effects individual patient-level meta-analysis which allows for assessment of effect modification (ie, interaction terms) by country or personal characteristics will be undertaken.[56] First, each individual study estimate will be calculated, plotted using a forest plot and compared with observe any similarities and differences. Next, individual estimates will be weighted using the Knapp-Hartung method and pooled using random effects methods taking into account both within and between variance as follows[57]:

### Patient and public involvement
There was no patient or public involvement in the design of the proposed study.

### Ethics and dissemination
Ethical approval was obtained from the following Human research Ethics Committees: Curtin University (HRECs) (ref No. HRE2022-0335), NSW Population and Health Services Research Ethics Committee (2022/ETH01392/2022.31), ACT Research Ethics and Governance Office approval under National Mutual Acceptance for multijurisdictional data linkage research (2022. STE.00208). For the ALSWH survey collections, ongoing ethical approval has beengranted by the HREC of the Universities of Newcastle and Queensland (approval numbers H076-0795 and 2004000224, respectively). The ALSWH also maintains institutional HREC approvals for external record linkage (approval numbers H-2011-0371 and 2012000132, respectively). In addition, access to national collections which include MBS/PBS, National

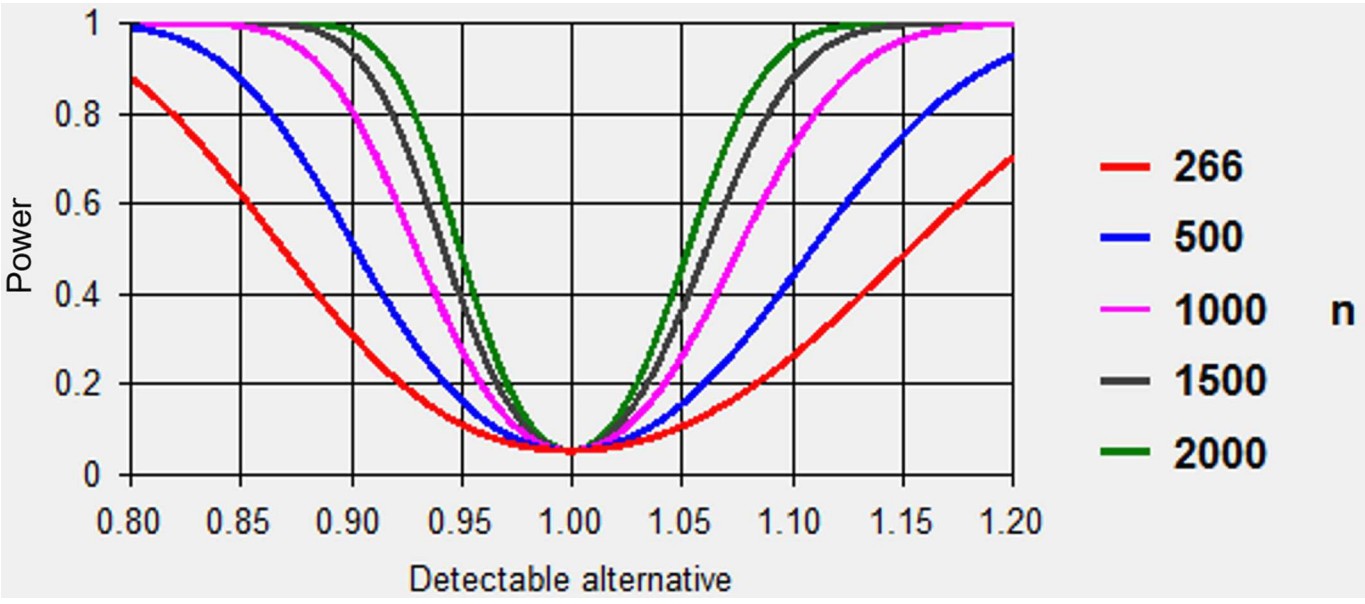

**Figure 3** Sample size calculation.

Death Index, Australian Cancer Database is approved by the Australian Institute of Health and Welfare. Access to state and territory data collections which include hospital admissions and cancer registries is approved by an appropriate HREC for each jurisdiction.[58] These data are approved by individual ethics committees and custodians in different jurisdictions, though the researchers make a single application to the ALSWH Data Access Committee, who then contact individual custodians on behalf of researchers. The Rotterdam Study data have been approved by the Medical Ethics Committee of the Erasmus MC (registration number MEC 02.1015) and by the Dutch Ministry of Health, Welfare and Sport (Population Screening Act WBO, license number 1071272–159521PG). The Rotterdam Study Personal Registration Data collection is filed with the Erasmus MC Data Protection Officer under registration number EMC1712001.

Each of the studies collected informed consent from all participants at recruitment. Women who participated in earlier waves of the ALSWH provided consent for access to administrative health data including the MBS, PBS, national death index, cancer registries and hospital records for most Australian States and Territories. In later waves, there has been a change to opt-out consent of these linkages, with participants provided information on the datasets that may be linked and a freecall phone number to opt out provided to participants at recruitment and in an annual newsletter. All study participants in the 45 and Up study consented to their survey data being linked to routinely collected health administrative data and health registry sources. All participants in the Rotterdam Study provided written informed consent to participate in the study and to have their information obtained from treating physicians.

Results of the proposed study will be published in high-impact journals and presented to key stakeholders via scientific meetings, lay press (print media, television, radio and internet).

## DISCUSSION

The equivocal literature regarding the potential association between CCB exposure and breast cancer is likely due to differences in study design (eg, cohort vs case control), length of follow-up, exposure measurement (eg, ever vs never or time duration) or population characteristics (eg, patients with hypertension and/or cardiovascular disease vs general population cohorts, which have differing risk of confounding by indication). These differences, along with small sample sizes in many studies (often n<1000), make inference difficult. Randomised, controlled trials are not usually suitable for identifying long-term adverse effects of medicine use and long-term monitoring through observational studies using administrative datasets has become the standard way of overcoming this limitation for postmarket surveillance research.[59] However, the sole use of administrative data limits the ability to control for critical lifestyle and other potential confounding factors

that are not routinely reported in such datasets, for example, BMI, physical activity, smoking and alcohol use history. In many studies, longitudinal cohorts have been used to alleviate this issue but often these studies have suffered from reporting and/or recall bias due to the reliance on self-report data, especially regarding the determination of exposure relating to prescription medication use. This has led to the preponderance of crude ever/never exposed to CCBs in the literature rather than more complex dose-duration measures that we propose.

Our study will involve six longitudinal cohorts of women extracted from internationally renowned Australian and Dutch studies linked with administrative health data, with complete coverage, to allow for continuous capture of CCB exposure for an extended period of time (ie, up to 19 years). Using state-of-the-art analytical techniques such as dose-duration methods previously unused in this area, this study will evaluate the association and characterise the dose–response nature of any such association between long-term CCB use and the risk of breast cancer. A wide range of person-related, temporal-related and geospatial level-related factors will be measured and adjusted to separate the effect of these factors from the relationship between CCB use and breast cancer. The analytic methods (non-linear threshold modelling) proposed are novel and will be a significant innovation to pharmacoepidemiological analyses, which currently lack such sophisticated methods.

Due to the nature of the observational study design, confounding by indication will be unavoidable. To minimise the impact of this issue, the study will be limited to participants with a diagnosis/self-reported hypertension and comparisons will include participants exposed to other prescribed (non-CCB) AHT medicines in addition to those who have no pharmacotherapy for hypertension. As explained in the methods, the Australian cohorts will be restricted by concessional status to capture the AHT exposure over the study duration, and our study is therefore likely prone to selection bias; this will be investigated in the sensitivity analyses. Complete ascertainment of prior exposure will not be possible since we will not have lifetime capture of prescribing data; therefore, CCB exposure may be under-reported. Due to the nature of medicine dispensing data, the level of certainty regarding whether the dispensed medicine (eg, combined hormonal contraceptives, combined menopausal hormone therapy and CCB/AHT medicines) was consumed by the patient is unclear. The diagnostic interval of breast cancer may vary between Australian and Dutch women, particularly since case ascertainment through cancer registries can take months from the initial diagnosis to coding notifications. Previous studies have shown a difference around 30 days in the diagnostic interval days depending on the presentation of symptoms and the temporal differences in diagnostic pathways between the two countries.[60] Another limitation is the absence of genetic data for this analysis. However, recent studies have indicated no association between genetic proxies for CCB or other

antihypertensives and breast cancer through Mendelian randomisation study design.[61 62]

While we have used the term 'association' here, our use of GPS adjustment provides a mechanism for undertaking analysis under a causal framework. Modelling each woman's propensity to receive the exposure as a function of covariates means that the GPS is independent of the outcomes and the study represents a quasi-experiment. We prefer to use the terminology 'association' when reporting our results so as not to overstate our ability to adjust for potential confounding. Interpretation of any association will incorporate the likelihood of causation (using the Bradford-Hill criteria).[63]

With regards to representativeness and generalisability of our results, leading epidemiologists[64] and reputable peer-reviewed publications[65 66] support the principle that cohort studies rely on the validity of internal comparisons to avoid systematic error and need not be a random sample of a larger population to provide generalisable knowledge. The validity of generalisation (ie, external not internal validity) depends on considerations about risk stratification and effect modification rather than bias/confounding. Thus, results of well-respected international cohort studies, such as the British Doctors' Study and US Nurses' Health Study, are regarded as providing generalisable knowledge despite their highly selected study samples.

The literature continues to debate whether or not CCBs increase the risk of breast cancer. The outcomes of the proposed study are likely to have considerable implications for both clinical practice and public health. If a positive association is found, this could have a global impact on clinical guidelines for treating hypertension in women. Long-term CCB use has the potential to become the number one major modifiable risk factor for breast cancer. Conversely if no association is found, this would provide reassurance to Australian and Dutch women that the current guidelines promoting CCBs as a first-line therapy for hypertension are appropriate and that taking a CCB does not pose additional breast cancer risk.

**Author affiliations**
[1]School of Population Health, Faculty of Health Sciences, Curtin University, Perth, Western Australia, Australia
[2]Cardiovascular Epidemiology Research Centre, School of Population and Global Health, The University of Western Australia, Perth, Western Australia, Australia
[3]Canberra Health Services, Canberra, Australian Capital Territory, Australia
[4]School of Medicine and Psychology, Australian National University, Canberra, Australian Capital Territory, Australia
[5]Institute for Health Research, The University of Notre Dame Australia, Fremantle, Western Australia, Australia
[6]Curtin Medical School, Curtin University, Perth, Western Australia, Australia
[7]PainChek, Sydney, New South Wales, Australia
[8]School of Public Health, The University of Queensland, Saint Lucia, Queensland, Australia
[9]School of Population Health, University of New South Wales, Sydney, New South Wales, Australia
[10]The NHMRC Medicines Intelligence Centre of Research Excellence, Sydney, New South Wales, Australia
[11]School of Population and Global Health, University of Western Australia, Perth, Western Australia, Australia
[12]Department of Epidemiology, Erasmus MC—University Medical Center, Rotterdam, Zuid-Holland, Netherlands
[13]Department of Internal Medicine, Maasstad Hospital, Rotterdam, Zuid-Holland, Netherlands
[14]Medical School, The University of Western Australia, Perth, Western Australia, Australia
[15]Melbourne Medical School, The University of Melbourne, Melbourne, Victoria, Australia
[16]The Royal Melbourne Hospital, Melbourne, Victoria, Australia
[17]Fiona Stanley Hospital, Perth, Western Australia, Australia
[18]School of Medicine, University of Tasmania, Hobart, Tasmania, Australia

**Contributors** CH assisted in the study design and drafted the manuscript. TNH and DY contributed to drafting sections relating to study design and methodology. RM, S-AP, DP, BS, CMS, WPA, CMR, JDH, MKB and DY conceptualised, designed the study, obtained the funding, reviewed and revised the manuscript. All authors revised the manuscript for important intellectual content. The author(s) read and approved the final manuscript.

**Competing interests** None declared.

**Patient and public involvement** Patients and/or the public were not involved in the design, or conduct, or reporting, or dissemination plans of this research.

**Patient consent for publication** Not applicable.

**Provenance and peer review** Commissioned; internally peer reviewed.

**ORCID iDs**
Chau Ho http://orcid.org/0000-0002-4285-9409
David Youens http://orcid.org/0000-0002-4296-4161
Max K Bulsara http://orcid.org/0000-0002-8033-6123
Jeffery David Hughes http://orcid.org/0000-0003-0040-4753
Bruno H Stricker http://orcid.org/0000-0003-3713-9762
Rachael Moorin http://orcid.org/0000-0001-8742-7151

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
