## [Reviewer comments · BMJ Open]

ARTICLE DETAILS

TITLE (PROVISIONAL)	Association between long-term use of calcium channel blockers (CCB) and the risk of breast cancer: A retrospective longitudinal observational study protocol
AUTHORS	Ho, Chau; Ha, Thi Ninh; Youens, David; Abhayaratna, Walter P.; Bulsara, Max; Hughes, Jeffery; Mishra, Gita; Pearson, Sallie-Anne; Preen, David; Reid, Christopher; Ruiten, Rikje; Saunders, Christobel M; Stricker, Bruno; van Rooij, Frank J.A; Wright, Cameron; Moorin, Rachael

VERSION 1 – REVIEW

REVIEWER	Jie Zhao University of Hong Kong
REVIEW RETURNED	26-Nov-2023

GENERAL COMMENTS	The authors aimed to examine the association of CCBs with breast cancer using multiple cohorts. It is an interesting question and the protocol has included detailed description. I have the following concerns for the authors to consider: 1. CCBs have different types, how the authors plan to deal with the heterogeneity?2. How the authors deal with the confounding? For example, the health status may be different for patients taking/not taking antihypertensive drugs.3. If the patients used combination of different antihypertensive drugs, how the authors deal with this situation?4. The cohorts are in different settings, is it possible that the use of CCBs is quite different in different cohorts? How the authors consider this issue?5. Mendelian randomization study is a study design which can minimize confounding, I would suggest the authors to include previous evidence on CCBs and cancer (https://pubmed.ncbi.nlm.nih.gov/37117874/; https://pubmed.ncbi.nlm.nih.gov/35113855/)
---

REVIEWER	Victoria Rotshild Hebrew University of Jerusalem
REVIEW RETURNED	04-Dec-2023

GENERAL COMMENTS	Introduction: 1. While citing several previously published articles, you omitted our study that was published in Cancers journal on May 2022. We evaluated a risk for breast cancer among CCB users. The exposure to CCBs was not associated with an increased risk of BCa (OR = 0.98; 95% CI, 0.92-1.04). Additionally, there was no association between long-term exposure to CCBs (above eight years) and
---

	increased BCa risk (OR = 0.91; 95% CI, 0.67-1.21). Higher cumulative doses of CCBs were not associated with an elevated risk of BCa (OR = 0.997; 95% CI, 0.962-1.034, calculated per 1000 DDD). I recommend citing our study as well. Methods  1. As reported in the Data Source, you plan to use the data from two Australian cohorts: the Australian Longitudinal Study on Women's Health (ALSWH) and the New South Wales (NSW) Sax Institute's 45 and Up Study (45 and Up Study). Please clarify how will you prevent duplication of the population. 2. Breast cancer is not restricted to females. I recommend including the male population as well in cohorts when available. 3. I recommend adding exclusion criteria of a personal history of any cancer diagnosed prior to cohort entry. 4. It is more appropriate to calculate a cumulative total dose in Defined Daily Dose (DDD) units instead of milligrams. Using DDDs allows combine different drugs from the same pharmacologic group
--	---

VERSION 1 – AUTHOR RESPONSE

Reviewer: 1

Dr. Jie Zhao, University of Hong Kong

Comments to the Author:

The authors aimed to examine the association of CCBs with breast cancer using multiple cohorts. It is an interesting question and the protocol has included detailed description.

I have the following concerns for the authors to consider:

1. CCBs have different types, how the authors plan to deal with the heterogeneity? As explained in the manuscript we recognise that different types of CCBs may have different impact on breast cancer risk and will examine this in subgroup analysis as mentioned in the 'Subgroup analysis' section "the analysis for each aim will be conducted separately according to CCB class (i.e., dihydropyridines and non-dihydropyridines) and duration of action (i.e., short or long action) where sufficient statistical power exists" (Page 15).

2. How the authors deal with the confounding? For example, the health status may be different for patients taking/not taking antihypertensive drugs. The strength of our study is that the data includes a wide range of socio-demographic, lifestyle and clinical potential confounders that may impact prescribing decisions for AHT medicines, a participant's decision to take prescribed AHT medicines or be associated with an increased risk of breast cancer. These confounding factors will be accounted for in the statistical models by using propensity scores as mentioned in the "Statistical analysis" section "The models will account for confounding factors at entry using propensity scores to balance these factors across exposure status captured over the follow-up period. In addition, we will use standard covariate adjustment for time varying covariates and any covariates that cannot be balanced using the propensity score. Any significant shifts in practice will be identified via the literature (and our clinical collaborators) and included as categorical variables in the models where appropriate" (Page 14).

3. If the patients used combination of different antihypertensive drugs, how the authors deal with this situation? As outlined in the 'Exposure ascertainment' section, "We will categorise AHT drugs dispensed into five main AHT drug classes: (i) beta-blocking agents (BB) (ATC code C07), (ii) CCB (C08); (iii) diuretics (C03), (iv) agents acting on the renin-angiotensin system (ACEIs and ARBs) (C09), or (v) other AHTs not included in these classes regardless of treatment therapies (i.e., monotherapy or combination therapy)" (Page 10-11). Subsequently, we will calculate the total cumulative duration exposed and the total cumulative dose of each AHT drug class in the combination, as specified in the 'Exposure ascertainment' section: "... the total cumulative duration

exposed and the total cumulative dose of each AHT drug class will be calculated from the medicine records available from entry to exit of the study" (Page 11). Thus, exposure to each drug class will be separately calculated and the model will therefore account for different combinations of exposure across classes.

4. The cohorts are in different settings, is it possible that the use of CCBs is quite different in different cohorts? How the authors consider this issue? The study incorporates two Australian cohorts and a Dutch cohort. To be eligible for analysis, participants from all three cohorts must satisfy identical inclusion and exclusion criteria, such as being alive and providing consent for linked data, demonstrating evidence of hypertension, having no history of breast cancer or bilateral mastectomy, and not using calcium channel blockers (CCB) prior to entry. Therefore, the basic eligibility criteria are identical across cohorts. The two Australian cohorts represent a single setting, as in Australia eligibility and access to prescribed medications is determined at the national level, so will not differ between the 45 and Up study participants (which is set in New South Wales only) and the ALSWH (which includes participants from across Australia). The reason we are including the two different cohort studies is because the baseline / follow-up surveys of each provides different information about the participants and this provides an opportunity to evaluate how this different information impacts the association of CCB use with breast cancer in the cohort specific analyses. The Rotterdam study represents a different setting and will allow us to (i) explore confounding and (ii) effect modification by setting related factors in the relationship between CCB use and breast cancer risk. Our study uses a distributed common analytical protocol methodology with a subsequent individual-level meta-analysis of the results. This study design has been used successfully in medication safety work where data from different jurisdictions cannot be brought together as it has been shown to improve replicability, transparency and minimize bias while increasing increase statistical power. As described in the 'Harmonisation of data' section, "To ensure content equivalence, each variable will be checked on (1) the definition used in the questionnaire, format, categories, unit, and time frame, (2) measurement method (e.g., self-reported, clinical examination, etc.), and (3) harmonisation rules" (Page 9-10). In the 'Study design' section, "Our analysis will: (i) first use identical protocols across cohorts (harmonised analyses) and then (ii) use variable/cohort-specific protocols to allow the influence of cohort-specific variables to be investigated (non-harmonised analyses)" (Page 6).

Reference:

Platt RW, Platt R, Brown JS, Henry DA, H. KO, Suissa S. How pharmacoepidemiology networks can manage distributed analyses to improve replicability and transparency and minimize bias. *Pharmacoepidemiol Drug Safety*. 2018;29(S1):3-7.

5. Mendelian randomization study is a study design which can minimize confounding, I would suggest the authors to include previous evidence on CCBs and cancer (<https://pubmed.ncbi.nlm.nih.gov/37117874/>; <https://pubmed.ncbi.nlm.nih.gov/35113855/>). We thank the reviewer for suggesting the reference and agree that this study design would be a useful addition. Unfortunately, we do not have access to genetic information of the participants and therefore cannot explore this option. While the Rotterdam Study has genome-wide data available, it is constrained to the inference of common variations and does not encompass rare mutations. Considering the expansive scale of larger cohort studies, there is a potential insufficiency in statistical power to effectively analyse this association in the current context. We have added the reference in the 'Discussion' section as tracked changes "Another limitation is the absence of genetic data for this analysis. However, recent studies have indicated no association between genetic proxies for CCB or other antihypertensives and breast cancer through Mendelian randomisation study design [60, 61]" (Page 19).

Reviewer: 2

Dr. Victoria Rotshild, Hebrew University of Jerusalem

Comments to the Author:

Introduction:

1. While citing several previously published articles, you omitted our study that was published in *Cancers* journal on May 2022. We evaluated a risk for breast cancer among CCB users. The exposure to CCBs was not associated with an increased risk of BCa (OR = 0.98; 95% CI, 0.92-1.04). Additionally, there was no association between long-term exposure to CCBs (above eight years) and increased BCa risk (OR = 0.91; 95% CI, 0.67-1.21). Higher cumulative doses of CCBs were not associated with an elevated risk of BCa (OR = 0.997; 95% CI, 0.962-1.034, calculated per 1000 DDD). I recommend citing our study as well. We thank the reviewer for suggesting the reference. We have added the reference as suggested in the 'Introduction' section "A similar pattern was observed in a recent study conducted by Rotschild et al (2022) in Israel, revealing an OR 0.997 (95% CI: 0.962-1.034) for a long-term exposure to CCBs (above 8 years) and breast cancer risk [19]" (Page 5).

Methods

1. As reported in the Data Source, you plan to use the data from two Australian cohorts: the Australian Longitudinal Study on Women's Health (ALSWH) and the New South Wales (NSW) Sax Institute's 45 and Up Study (45 and Up Study). Please clarify how will you prevent duplication of the population.

We appreciate the reviewer for bringing our attention to this concern. Identifying potential duplication between the two groups is challenging since both the 45 and Up Study and ALSWH anonymize participant information and do not ask participants if they are or have participated in any other cohort studies. We are also directly prevented from undertaking a direct review of participation overlap by trying to identify participants who have matching criteria across the cohorts as we have signed agreements with the data custodians of each cohort not to look at the two cohorts side by side – in fact the cohorts must be analysed by different study personnel to prevent any attempt to cross reference participants. To estimate the maximum potential duplication, we determined (at an aggregated level) the number of ALSWH participants residing in NSW close to the baseline year of the 45 and Up Study (2006). Additionally, we considered the number of 45 and Up participants within the birth year range of the two ALSWH cohorts (1921-26 and 1946-51), as illustrated in the table below.

Year of birth 1921-26 Year of birth 1946-51

Female ALSWH participants residing in NSW n=2,490 n=3,182

Female 45 and Up participants residing in NSW n=7,724 n=28,596

This analysis shows that it is possible that these NSW participants in the ALSWH cohort could also be included in the 45 and Up Study. Unfortunately, due to sample size issues with the ALSWH we feel it would be detrimental to exclude the NSW from the base analysis. However to evaluate the impact of this potential duplication in a sensitivity analysis, we will systematically exclude ALSWH participants residing in NSW to assess the impact on the study results due to potential duplication with those enrolled in the 45 and Up Study. The following sentence has been incorporated into the 'Sensitivity analysis' section: "Due to the potential duplication of participants who were part of the ALSWH and also enrolled in the 45 and Up Study residing in NSW, we will systematically exclude these ALSWH participants in a sensitivity analysis. This exclusion is essential for evaluating the potential impact of duplication on the study results" (Page 16).

References:

https://alswh.org.au/wp-content/uploads/2020/01/Old_4databook.pdf

https://alswh.org.au/wp-content/uploads/2020/01/Mid_4databook.pdf

https://www.saxinstitute.org.au/wp-content/uploads/December-2011-DATA-BOOK-_final_5531.pdf

2. Breast cancer is not restricted to females. I recommend including the male population as well in cohorts when available.

We appreciate the reviewer's suggestion. While we acknowledge that men can be diagnosed with breast cancer, it constitutes less than 1% of all breast cancer cases. Male breast cancer exhibits distinct hormone expression, environmental risk factors, and mechanisms. Given these

considerations our clinical collaborators have confirmed that we should maintain our original restriction to the female population for this analysis.

References:

<https://www.cancer.org.au/cancer-information/types-of-cancer/breast-cancer-in-men>

Anderson WF, Jatoi I, Tse J, Rosenberg PS. Male breast cancer: a population-based comparison with female breast cancer. *Journal of Clinical Oncology*. 2010 Jan 1;28(2):232.

Fentiman IS, Fourquet A, Hortobagyi GN. Male breast cancer. *The Lancet*. 2006 Feb 18;367(9510):595-604.

3. I recommend adding exclusion criteria of a personal history of any cancer diagnosed prior to cohort entry.

We thank the reviewer for the recommendation. As we are also interested in examining the effect of a personal history of any cancer at cohort entry and the risk of breast cancer and CCB, we have added the sentence as tracked changes in the 'subgroup analysis' section in the manuscript "Moreover, we will conduct a subgroup analysis to assess the influence of a personal history of any cancer at cohort entry on the risk of breast cancer and CCB" (Page 15).

4. It is more appropriate to calculate a cumulative total dose in Defined Daily Dose (DDD) units instead of milligrams. Using DDDs allows combine different drugs from the same pharmacologic group.

In terms of the availability of data such as the date of dispensing, quantity dispensed, and the strength of the medication in the medication data, we can employ milligrams to determine the total cumulative dose of each antihypertensive drug that participants were actually exposed to. This is our preference for the base analysis since the Defined Daily Dose represents the assumed average maintenance dose per day, rather than the actual dose of CCB the patient was exposed to. We recognise that DDDs were developed by the WHO in order to facilitate the examination of changes in drug use over time, allow international comparisons, allow measurement of the impact of an intervention on drug use, document the relative therapy intensity, follow changes in the use of classes of drugs and evaluate regulatory effects etc. on prescribing patterns. DDDs were not designed for evaluation of the impact of dose on outcomes per se. As our interest is in the possible dose response of exposure to the amount of CCB on the risk of breast cancer and we are going to great lengths to model this as accurately as possible we prefer to use the actual measure of exposure rather than the average maintenance dose per day as represented by the DDD. We did however state in the manuscript that we will use DDD instead of actual dose in a sensitivity analysis so that comparison of our results can be made with existing studies, detailed in the 'sensitivity analysis' section: "The potential impact of reduced exposure resulting from imperfect adherence and prescribing at defined daily dose levels (a statistical measure of drug consumption, defined by the World Health Organization) will be investigated in the dose-duration models and through sensitivity analyses [36]" (Page 15-16).